# Surface-acoustic-wave-induced
# unconventional superconducting pairing

**Viktoriia Kornich**

Institut für Theoretische Physik und Astrophysik, Universität Würzburg,
97074 Würzburg, Germany
Kavli Institute of Nanoscience, Delft University of Technology,
2628 CJ Delft, The Netherlands

viktoriia.kornich@physik.uni-wuerzburg.de

## Abstract

Unconventional superconductivity is usually associated with symmetry breaking in the system. Here we consider a simple setup consisting of a solid state material with conduction electrons and an applied surface acoustic wave (SAW), that can break time and spatial translation symmetries. We study the symmetries of the possible SAW-induced order parameters, showing that even-frequency spin-triplet odd-parity order parameter can occur. We suggest different methods of how to engineer the symmetries of the order parameters using SAWs and the applications of such setups.

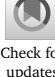

# 1   Introduction

Unconventional superconductivity is a quickly-developing field, studying new and interesting features of superconducting materials and setups [1–6]. These can be, for example, high-$T_c$ superconductors [7], spin-triplet superconductors [8], nematic superconductors [9]. Due to the wide variety of unusual properties, there is no precise definition of an unconventional superconductor. However a conventional superconductor is usually defined as [10] the one with s-wave Cooper pairs formed due to phonon-mediated electron-electron attraction as described in BCS theory [11]. One of the known benefits of discovering new, unconventional, properties of the materials, that reveal themselves strongly in a superconducting state, is that they can provide information about their normal state [10]. For example, it is widely believed that high-$T_c$ superconductivity is due to antiferromagnetic spin-fluctuation electron-electron interaction [6]. While it is a matter of the ongoing research, which is not in the scope of this paper, specifics of spin structure and electronic correlations in materials are definitely of high interest outside of the field of superconductivity.

To understand the underlying mechanisms of unconventional superconductivity better, to make their properties more controllable or even to engineer such superconductors, it can be useful to study the problem from the other side, i.e. try to reproduce their properties in some nanodevices. The well-known example of such approach, that has been followed by the number of theoretical and experimental works, see e.g. [12–15], is p-wave superconductivity in a nanowire-superconductor nanostructure [16, 17]. In this view, it would be interesting to consider simple devices, where certain symmetries are broken that induce unconventional superconducting pairings. Certainly, the pairing type that is most beneficial energetically will be dominating.

Modern experimental techniques allow for measuring non-equilibrium superconducting states, [18, 19] and the effects of nanostrain and SAWs on the existing superconductivity [20, 21]. We propose to induce non-equilibrium unconventional superconducting pairing via externally applied SAW and investigate how the time and spatial shape of the SAW affects the type of induced superconducting pairing.

In this work, we consider a device, consisting of a solid state material and an induced surface acoustic wave [22] on it. SAW induces electron-electron interaction in analogy to phonon-mediated electron-electron interaction in conventional superconductors. However the mechanism is different. SAW is not a heat bath, it is produced by an external source, so it can be understood as forcing electrons to interact. Consequently, the SAW-induced interaction has distinct features coming from the shape of the SAW, different frequency and momentum dependence, and thus different types of Cooper pairs can be induced. Most importantly, due to time and spatial translation symmetries breaking SAW-induced order parameters can have various symmetries, e.g. even-frequency spin-triplet odd-parity type, to which we refer as unconventional superconducting pairing. The conditions for the actual superconducting phase transition require detailed and complex analysis taking into account many parameters, e.g. electron density of states, other sources of electron-electron interactions, such as Coulomb interaction. We discuss some of these points in the end of our paper, however this is beyond the scope of this paper as our aim is to show that SAW-induced superconducting electron pairing can have various symmetries, but not to study a superconducting state of matter or the conditions for the superconducting transition. We plan to perform these studies in our future work.

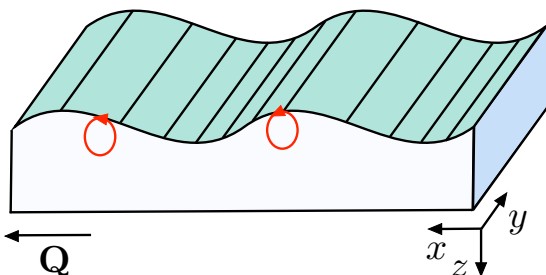

Figure 1: Rayleigh surface acoustic wave propagating with the wave vector **Q** parallel to the axis $x$. The axis $z$ is the coordinate of the depth of the material. Atoms on the surface have elliptical trajectories, which is a combination of longitudinal and transverse modes [25].

## 2 Electron Dynamics

In this section, we consider electrons and their dynamics with a time-dependent electron-electron interaction, without specifying its origin. The action of electrons is $S_e = S_0 + S_{e-e}$, where $S_0$ includes their kinetic energy and $S_{e-e}$ describes electron-electron interaction:

$$S_0 = \int dt d\mathbf{r} \, \bar{\psi}_{t,\mathbf{r}} \left( i\partial_t + \frac{\nabla^2}{2m} + \mu \right) \psi_{t,\mathbf{r}}, \tag{1}$$

$$S_{e-e} = \int \prod_{i=1}^{2} dt_i d\mathbf{r}_i \, \bar{\psi}_{t_1,\mathbf{r}_1} \bar{\psi}_{t_2,\mathbf{r}_2} W_{\mathbf{r}_1,\mathbf{r}_2,t_1,t_2} \psi_{t_2,\mathbf{r}_2} \psi_{t_1,\mathbf{r}_1}, \tag{2}$$

where $t$ is time, $\mathbf{r}$ is the coordinate in 3D space, $m$ is the electron mass, $\psi$ and $\bar{\psi}$ are electron Grassmann fields, and $W$ is their interaction potential. The electron-electron interaction contains four electron Grassmann fields [23], which can be decoupled via Hubbard-Stratonovich transformation in a Cooper channel with a bosonic field $\phi$. In short, Hubbard-Stratonovich transformation is a multiplication of a partition function $\mathcal{Z}_e = \mathcal{Z}_0^e \int \mathcal{D}(\psi, \bar{\psi}) \exp[iS_e]$ with an extended unity, which allows to cancel the term with four fields, in this case $\psi$ and $\bar{\psi}$, and instead obtain the terms of the type $\bar{\phi} W^{-1} \phi$ and $\bar{\psi}\bar{\psi}\phi$, $\bar{\phi}\psi\psi$. However this also adds a separate action for the introduced bosonic field. Thus we obtain

$$
\tilde{S}_{e-e} + S_\phi = \int \prod_{i=1}^{2} dt_i d\mathbf{r}_i \sum_{\sigma,\sigma'} [\bar{\psi}^\sigma_{t_1,\mathbf{r}_1} \bar{\psi}^{\sigma'}_{t_2,\mathbf{r}_2} \phi^{\sigma'\sigma}_{\mathbf{r}_2,\mathbf{r}_1,t_2,t_1} \tag{3}
$$
$$
+ \bar{\phi}^{\sigma\sigma'}_{\mathbf{r}_1,\mathbf{r}_2,t_1,t_2} \psi^{\sigma'}_{t_2,\mathbf{r}_2} \psi^\sigma_{t_1,\mathbf{r}_1} - \bar{\phi}^{\sigma\sigma'}_{\mathbf{r}_1,\mathbf{r}_2,t_1,t_2} \left( W^{-1} \right)^{\sigma\sigma'}_{\mathbf{r}_1,\mathbf{r}_2,t_1,t_2} \phi^{\sigma'\sigma}_{\mathbf{r}_2,\mathbf{r}_1,t_2,t_1} ].
$$

Here the last term in the square brackets belongs to $S_\phi$, and $W^{-1}$ is the inverse of the electron-electron interaction potential. We have added spin indices $\sigma$ and $\sigma'$, because fermions obey Pauli principle and thus the spin part of the superconducting pairing is important [10].

In order to characterize electrons in a dynamic, time-dependent system, we need to use non-equilibrium Keldysh formalism. After we transfer to the Keldysh contour time representation, i.e. introduce indices for its upper and lower branches, 1 and 2, respectively, we perform integration over $\psi$ and $\bar{\psi}$ in $\mathcal{Z}_e$, thus keeping only $\phi$ and $\bar{\phi}$ degrees of freedom. Then we apply Keldysh rotation with the transformation matrix $U_f = \frac{1}{\sqrt{2}} \begin{pmatrix} 1 & -1 \\ 1 & 1 \end{pmatrix}$ and obtain the effective

action for bosonic fields:

$$S_{\text{eff}} = S_\phi^K - i\text{Tr}\ln\left[(i\partial_t + \left[\frac{\nabla^2}{2m} + \mu\right]\tau_z + \phi_c)\gamma_0 + \phi_q\gamma_x\right], \tag{4}$$

$$S_\phi^K = -2\int\prod_{i=1}^2 dt_i d\mathbf{r}_i\,(\bar{\phi}_c \quad \bar{\phi}_q)(W^K)^{-1}\begin{pmatrix}\phi_c\\\phi_q\end{pmatrix}. \tag{5}$$

Here $\tau_z$ is a Pauli matrix in a particle-hole space, and $\gamma_{0,x}$ are Pauli matrices in Keldysh space. The bosonic fields $\phi_c$ and $\phi_q$ are classical and quantum components of $\phi$, i.e. $\phi_c = (\phi_1+\phi_2)/2$ and $\phi_q = (\phi_1-\phi_2)/2$, where indices 1 and 2 denote branches of the Keldysh contour. Analogously, we define $(W^K)^{-1} = W_q^{-1}\gamma_0 + W_c^{-1}\gamma_x$. At this point, we would like to remind that $\phi_c$ and $\phi_q$ are 4x4 matrices in spin and particle-hole spaces. We have also omitted time and coordinate indices for brevity of notation in Eq. (4). To obtain Eq. (5), we used bosonic transformation for Keldysh rotation, $U_b = \frac{1}{\sqrt{2}}\begin{pmatrix}1 & 1\\1 & -1\end{pmatrix}$.

The argument of the logarithm in Eq. (4) can be denoted as an inverse electron Green's function, $G^{-1} = G_0^{-1} + \phi_c\gamma_0 + \phi_q\gamma_x$. Thus Gor'kov equation has the form $G^{-1}G(t_1, t_2, \mathbf{r}_1, \mathbf{r}_2) = \delta(t_1-t_2)\delta(\mathbf{r}_1-\mathbf{r}_2)$. If we assume that $G$ and $\phi_c$ depend only on the differences of times and coordinates and consider mean-field approximation, i.e. $\phi_q = 0$, the solution for the retarded and advanced Green's functions is

$$G^{R(A)} = [\omega - \xi_k\tau_z + \phi_c \pm i0]^{-1}, \tag{6}$$

where, $\xi_k = k^2/(2m) - \mu$. These Green's functions have a structure $G = \begin{pmatrix}\mathcal{G} & F\\\bar{F} & \bar{\mathcal{G}}\end{pmatrix}$, where $\mathcal{G}$ and $\bar{\mathcal{G}}$ are particle and hole Green's functions, and $F$ and $\bar{F}$ are anomalous Green's functions. Anomalous Green's functions characterize the type of the superconductivity, e.g. odd- or even-frequency, spin-singlet or spin-triplet, parity. In our case the expressions for the anomalous Green's functions are rather bulky, therefore we do not present them here, however if we neglect all higher order terms of $\phi_c$ except for the linear ones, we obtain a simple relation between anomalous Green's functions and bosonic fields $\phi_c$, namely $F \propto \phi_c$. We note that we took this limit at the moment only for the purposes of demonstration of the dependence between $F$ and $\phi_c$ as the relation between $\xi_k$, $\omega$, and $\phi_c$ can be different. However it is clear that in order to find the symmetries of $F$, we need to find the symmetries of $\phi_c$. In principle, finding characteristics of $\phi_c$ can be sufficient in order to define certain types of superconductivity, e.g. even-frequency, spin-triplet, p-wave superconductor.

To find the symmetry of bosonic fields, $\phi_c$, we derive the system of equations for the stationary phase solutions [23]. Each of them is a variation of the action with respect to the desired bosonic field, e.g. $\delta S_{\text{eff}}/\delta\phi_c^{\uparrow\uparrow} = 0$. These equations are rather bulky in our model, therefore we simplify the equation assuming $\phi_c \ll \xi_k, \omega$. Thus we obtain

$$\bar{\phi}_{c,\omega',\mathbf{k}'}^{\uparrow\uparrow} = \int d\omega d\mathbf{k}\frac{W_{q,\omega,\omega',\mathbf{k},\mathbf{k}'}^{\uparrow\uparrow,K}\bar{\phi}_{c,\omega,\mathbf{k}}^{\uparrow\uparrow}}{2i(\xi_k^2 - \omega^2 + |\phi_{c,\omega,\mathbf{k}}^{\downarrow\downarrow}|^2 + |\phi_{c,\omega,\mathbf{k}}^{\uparrow\uparrow}|^2 + \bar{\phi}_{c,\omega,\mathbf{k}}^{\uparrow\downarrow}\phi_{c,\omega,\mathbf{k}}^{\downarrow\uparrow} + \bar{\phi}_{c,\omega,\mathbf{k}}^{\downarrow\uparrow}\phi_{c,\omega,\mathbf{k}}^{\uparrow\downarrow})}. \tag{7}$$

This equation is similar to the well-known equation for the order parameter in BCS formalism [11]. The difference is the presence of different types of mean fields in the denominator. If we neglect all $\phi_c$ terms in the denominator, we will obtain a rather simple equation, which can already give us a symmetry of $\phi_c^{\uparrow\uparrow}$. If $W_{q,\omega,\omega',-\mathbf{k},\mathbf{k}'}^{\uparrow\uparrow,K} = -W_{q,\omega,\omega',\mathbf{k},\mathbf{k}'}^{\uparrow\uparrow,K}$, then in order for Eq. (7) to hold, $\phi_{c,\omega,-\mathbf{k}}^{\uparrow\uparrow} = -\phi_{c,\omega,\mathbf{k}}^{\uparrow\uparrow}$, i.e. the mean field is odd in momentum and thus can be a p-wave

superconducting pairing. Analogously, there can be other symmetries, depending on electron-electron interaction form. At this point we finish consideration of the electron dynamics, as we showed how the superconducting pairing depends on the time-dependent electron-electron interaction and how anomalous Green's function can be obtained.

## 3 SAW-Induced Electron-electron Interaction

### 3.1 Mechanism and General Description

In this section, we consider the mechanism for the electron-electron interaction in detail. The SAW is induced externally and has certain energy, which can be described as a sum of energies of oscillating quanta, analogous to thermal phonons, $E_{\mathrm{w}} = \sum_k E_k^{\mathrm{ph}}$. Once an electron passes through the SAW, it disturbs its charge distribution. Then the other electron will pass through this SAW with a modified charge distribution. Thus the SAW transmits information from one electron to the other, and in such a way induces their interaction. If the electrons pass through the SAW quicker than the electron-phonon scattering time, $\tau_{\mathrm{el-ph}}$, there can be a virtual process, when the first electron absorbs a phonon, and the second one emits the same phonon. Consequently, energy is conserved, $E(\mathbf{p_1}) + E(\mathbf{p_2}) + E_{\mathrm{w}} = E(\mathbf{p_1'}) + E(\mathbf{p_2'}) + E_{\mathrm{w}}$, where $E$ is energy of an electron and $\mathbf{p_1} + \mathbf{k} = \mathbf{p_1'}$ and $\mathbf{p_2} - \mathbf{k} = \mathbf{p_2'}$. If $\mathbf{p_1} = -\mathbf{p_2}$ then $\mathbf{p_1'} = -\mathbf{p_2'}$ and we obtain superconducting pairing with opposite momenta, see Fig. 2. The difference to the conventional superconducting pairing is that there $E_{\mathrm{w}} = 0$, and the first electron emits a phonon which is absorbed by the second one.

At longer times there will be various processes of emission and absorption of phonons by electrons. This might lead to non-equilibrium quasiparticle distribution function and processes enhancing superconductivity or reducing it (see e.g. Ref. [24]). In this work, we focus on electron pairing symmetry and not on conditions for superconducting phase transition, therefore we do not discuss these effects further here.

The derivation of the electron-electron interaction mediated by an externally-induced acoustic wave is also described in terms of Keldysh Green's functions. In general form, the action for electron-wave interaction is

$$S_{\mathrm{e-w}} = \int V_{\mathbf{k}}[a_{t,\mathbf{k}} - \bar{a}_{t,-\mathbf{k}}]\bar{\psi}_{t,\mathbf{k+K}}\psi_{t,\mathbf{K}} dt\, d\mathbf{k}\, d\mathbf{K}, \tag{8}$$

where bosonic fields $a$ describe quanta of the oscillations of the wave, analogously to phonons, but here they have externally defined directions and magnitudes of $\mathbf{k}$. The interaction potential $V(\mathbf{k})$ describes the electron-wave interaction and characterizes the shape of the wave. For

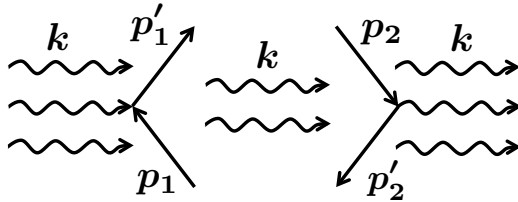

Figure 2: The pairing mechanism under consideration: the first electron absorbs a phonon from the wave, thus the second electron interacts with a modified wave and emits the same phonon. This virtual energy-conserving process can occur at times $\ll \tau_{\mathrm{el-ph}}$.

example, in case of a single plane wave with momentum $\mathbf{Q}$, $V(\mathbf{k}) = V_0(\mathbf{k})\delta(\mathbf{k} - \mathbf{Q})$, where $V_0(\mathbf{k})$ depends on the symmetries of the crystal, intensity of the wave, and physical process corresponding to electron-wave interaction, e.g. piezoelectric effect or deformation of the band structure. If it is a standing wave, then $V(\mathbf{k}) = V_0(\mathbf{k})(\delta(\mathbf{k} - \mathbf{Q}) + \delta(\mathbf{k} + \mathbf{Q}))$. If the wave has some more complex structure, that is many harmonics, then $V(\mathbf{k}) = \sum_{\mathbf{Q}} V_{cs}(\mathbf{k})\delta(\mathbf{Q} - \mathbf{k})$ with Fourier coefficients $V_{cs}(\mathbf{k})$ depending on both the wave shape and the crystal characteristics. Here, the difference to the thermal phonons becomes obvious: the contributions to $V(\mathbf{k})$ come only with particular wave vectors $\mathbf{Q}$, that is break spatial translation symmetry of the crystal.

In order to obtain $W^K$ and correctly take into account non-equilibrium dynamics of $a$, we transfer to Keldysh space and perform Keldysh rotation as $\Psi_{t,\mathbf{K}} = U_f \gamma_z \begin{pmatrix} \psi_{1,t,\mathbf{K}} & \psi_{2,t,\mathbf{K}} \end{pmatrix}^T$, $\bar{\Psi}_{t,\mathbf{k}+\mathbf{K}} = \begin{pmatrix} \bar{\psi}_{1,t,\mathbf{k}+\mathbf{K}} & \bar{\psi}_{2,t,\mathbf{k}+\mathbf{K}} \end{pmatrix} U_f^{-1}$. The additional index 1 or 2 for $\psi$ and $\bar{\psi}$ denotes branches of the Keldysh contour: 1 is for the upper branch of the Keldysh contour and 2 is for the lower one.

Then we derive effective electron-electron interaction mediated by the wave in a following way. The partition function of the system is by definition $\mathcal{Z} = \mathcal{Z}_0 \int \mathcal{D}(\bar{\psi}, \psi, \bar{a}, a) \exp[iS]$, where $S = S_0 + S_{e-w} + S_w$ is the full action of the system with $S_w$ describing the applied wave. We expand exponent in $S_{e-w}$, perform field integration over $\bar{a}$ and $a$, and assume that all odd terms of expansion turn to zero analogously to conventional phonon case [23]. After collecting all non-zero terms of the expansion, we obtain $\mathcal{Z} = \int \mathcal{D}(\bar{\psi}, \psi) \exp[iS_0 - \langle S_{e-w} S_{e-w} \rangle / 2]$. The last term in the exponent gives effective interaction of electrons mediated by the external wave, which can be expressed as

$$S_{e-e} = \frac{i}{2}\langle S_{int} S_{int} \rangle = \int \prod_{i=1}^{2} dt_i d\mathbf{r}_i W^K_{t_1, t_2, \mathbf{r}_1, \mathbf{r}_2} \bar{\Psi}_{t_1, \mathbf{r}_1} \Psi_{t_1, \mathbf{r}_1} \bar{\Psi}_{t_2, \mathbf{r}_2} \Psi_{t_2, \mathbf{r}_2}, \tag{9}$$

where electron-electron interaction potential in Fourier representation is

$$W^K_{\omega_1, \omega_2, \mathbf{k}_1, \mathbf{k}_2} = \frac{i}{2} V_{\mathbf{k}_1} V_{\mathbf{k}_2}[D_{\omega_1, \omega_2, \mathbf{k}_1, -\mathbf{k}_2} - D_{\omega_2, \omega_1, \mathbf{k}_2, -\mathbf{k}_1}]. \tag{10}$$

Here matrices $D = \begin{pmatrix} D_K & D_A + D_R \\ D_A + D_R & D_K \end{pmatrix}$ include Keldysh, advanced, and retarded Green's functions for the wave bosonic operators $a$: $D_K$, $D_A$, and $D_R$, respectively.

In many cases, $D_A$, $D_R$, and $D_K$ do not depend on the directions of $\mathbf{k}_1$ or $\mathbf{k}_2$. Therefore the change of sign of $W^K_{\omega_1, \omega_2, \mathbf{k}_1, \mathbf{k}_2}$ due to the change of sign of the momentum can come from $V_{\mathbf{k}_1}$ or $V_{\mathbf{k}_2}$. Namely, if the electron-wave interaction is different in different directions. This is possible in crystals without inversion symmetry, e.g. in piezoelectric materials.

## 3.2 Rayleigh Surface Acoustic Wave

For demonstration of the principle discussed in general above, let's consider a particular example of a surface acoustic wave, the Rayleigh SAW. Let's assume, it is propagating in a slab of a solid state material in the $x$ direction, the direction perpendicular to the surface of the slab is $z$, see Fig. 1. For this type of SAW, the atoms on the surface perform elliptic motion, thus this type of wave contains both longitudinal and transverse components and the displacement operator is

$$\mathbf{u} = \sqrt{\frac{\hbar}{2M\Omega}} \mathbf{e}(\mathbf{Q}) e^{-Qz} e^{i\mathbf{Q}\cdot\mathbf{x}} (a_Q e^{-i\Omega t} - a^\dagger_{-Q} e^{i\Omega t}), \tag{11}$$

$$\mp e_x(\pm Q) = i e_z(Q) = (A\Omega)^{1/2}, \tag{12}$$

where $\mathbf{Q}$ is the wave vector parallel to the axis $x$, $\Omega$ is the frequency of the wave, $M$ is the mass of an atom of the lattice, and $\mathbf{e}$ is the polarization vector, which in this case, has two components, $e_x$ and $e_z$. $A$ is a normalization constant that does not depend on $Q$ or $\Omega$ for the linear spectrum. In principle, as the sound velocities are usually different for longitudinal and transverse modes, $e_x$ and $e_z$ usually have two terms with the exponential decay proportional not only to $Q$, but also to the prefactors $\leq 1$ depending on the ratio between the given mode velocity and the Rayleigh wave velocity [25]. Here we neglect these details for the shortness of notation.

We note that the applied SAW breaks spatial translation symmetry in the system as follows from Eqs. (11) and (12). Symmetry breaking in the system can induce different types of pairing and their co-existence [26–29], e.g. odd-frequency spin-singlet odd-parity pairing [30, 31].

### 3.3 Rayleigh SAW-induced Electron-electron Interaction in Piezoelectric Material

Let's assume, the standing Rayleigh SAW is induced in a thin piezoelectric slab, so that we can neglect the decay of the wave in depth of the material, i.e. the term $e^{-Qz}$ in Eq. (11). The material polarizes and produces electric field that interacts with electrons. Such electric field in the bulk of a piezoelectric material is usually described as $E_i = -h_{ikj}\varepsilon_{kj}/\epsilon_r$, where $\epsilon_r$ is a dielectric constant, $h_{ikj}$ is a piezoelectric tensor, and $\varepsilon_{kj}$ is a strain tensor, $\varepsilon_{kj} = \frac{1}{2}\left(\frac{\partial u_k}{\partial x_j} + \frac{\partial u_j}{\partial x_k}\right)$. Then the electron-wave interaction is the interaction of the induced electric potential $\Phi(\mathbf{r})$ and the electron density $\rho(\mathbf{r}) = \psi^\dagger(\mathbf{r})\psi(\mathbf{r})$, namely $H_{e-w} = -e\int \rho(\mathbf{r})\Phi(\mathbf{r})d\mathbf{r}$. This brings us to the action describing electron-wave interaction, Eq. (8).

Now let's assume some particular material and check, whether our model gives an interesting result there. The widely-used material with piezoelectric properties is GaAs. Its piezoelectric tensor is $h_{ikj} = h_{14}|\epsilon_{ikj}|$, where $\epsilon_{ikj}$ is a Levi-Civita symbol with indices corresponding to the main crystallographic axes. This gives $\Phi_{\text{GaAs}} = -2h_{14}\varepsilon_{xz}L_y/\epsilon_r$, where $L_y$ is the length of the sample in the $y$ direction.

Let's consider $\Phi_{\text{GaAs}}$ in more detail. The term $\partial u_z/\partial x$ of the strain tensor component $\varepsilon_{xz}$ changes sign with $\mathbf{Q}$, because the differentiation over $x$ gives a prefactor $\pm Q$ and $e_z$ does not depend on the sign of $\mathbf{Q}$, see Eqs. (11) and (12). In such a way, $\Phi_{\text{GaAs}}(-\mathbf{Q}) = -\Phi_{\text{GaAs}}(\mathbf{Q})$ and consequently $V_{-\mathbf{Q}} = -V_{\mathbf{Q}}$. This means that $W^K_{\omega_1,\omega_2,-\mathbf{k}_1,\mathbf{k}_2} = -W^K_{\omega_1,\omega_2,\mathbf{k}_1,\mathbf{k}_2}$, and the bosonic field $\phi_c^{\uparrow\uparrow}$ has a node and is odd in momentum, consequently this system satisfies the necessary condition for the spin-triplet, p-wave superconducting electron pairing. We note, that the ideal propagating wave would not lead to such effect, as we obtain $W^K_{\omega_1,\omega_2,\mathbf{Q},\mathbf{Q}}$ in that case. However, in real experiment the induced wave will be most likely of a complex shape and thus will not induce such electron-electron interaction even in a propagating case.

## 4 Discussion and Conclusions

We would like to point out that there are other mean fields in our model, e.g. $\phi_c^{\uparrow\downarrow}$ that can lead to spin-singlet superconducting pairing. Also, if the crystal does not have inversion symmetry, there can be spin orbit coupling, and this can lead to a state with mixed parity. We do not include such effects in this paper for shortness of notation. However, depending on the relevance of them, this can be necessary in order to study the actual induction of superconductivity. The stationary phase equations, as Eq. (7), can have multiple solutions including zero solution [23]. In case of Eq. (7), it can potentially be, for example, $f$-wave instead of

$p$-wave parity. The co-existence of different phases and order parameters must be considered, if the superconducting state is investigated.

The Rayleigh SAW can be induced on a piezoelectric substrate via an interdigital transducer [32, 33]. SAWs can also be induced via laser pulses [34], thus allowing for non-piezoelectric materials in our setup. Electrons in an appropriate material must be strongly susceptible to the applied acoustic wave. One possible type of such materials is flat-band systems [35, 36]. Due to the infinite electron density of states they can support surface superconductivity with critical temperatures much higher than in the bulk [37]. There are also metamaterials with various acoustic properties [38–41], and depending on the other parameters needed to induce superconductivity, e.g. temperature and electron density of states, they might also appear to be applicable to our scheme.

In order to obtain superconducting state, the wave-induced electron-electron interaction must be correctly balanced with the Coulomb interaction. In case of conventional superconductivity, the derivation of phonon-mediated electron-electron coupling constant and Coulomb pseudopotential were derived in Ref. [42]. These quantities can be compared and give information about relative strengths of these two types of electron-electron interaction. Coulomb interaction is not necessarily destructive for superconducting phase. For example, it was shown theoretically that short-range Coulomb interaction can make odd-parity order of superconductivity dominant over s-wave order [43–45], because electrons in spin-triplet odd-parity pairing are more spatially separated than in case of spin-singlet s-wave pairing. The strength of Coulomb interaction can be varied e.g. by the applied metallic gates, and thus can even serve as a means to control and regulate the type of superconductivity. More detailed study of material properties of certain oxide heterostructures showed that the odd-parity superconducting order can be stabilized by the orbital effects of the material [46]. The stabilization of superconductivity in a real material is a complex task, see e.g. Refs. [47–49] on theoretical studies of effects in $SrTiO_3$ due to softening of transversal phonon mode near the quantum critical point.

In this work, we showed that the shape of the applied SAW can change the symmetry of the order parameters, because it determines how momentum $\mathbf{Q}$ enters the electron-electron interaction, and most importantly, how it breaks spatial symmetry of the crystal. This shows a possible way to externally define the symmetry of an induced superconducting pairing. In this system, for example, we can obtain $p$-wave pairing of Cooper pairs. The $p$-wave superconductors are very often associated with Majorana bound states [50]. If it becomes possible to obtain $p$-wave superconductor in any direction, that is in the direction of the applied SAW, it could assist efforts to perform braiding of Majoranas. Even more generally, defined superconducting paths with defined symmetries can be a low-decoherence way to induce interactions between qubits. Such paths do not have to be straight, but their shape must be taken into account as it breaks spatial translation symmetry. Apart from that the co-existence of several types of order parameters must be taken into account.

Applied SAW can also break time-related symmetries. In our example for Rayleigh SAW we focused on the spatial translation symmetry breaking and did not discuss the possible induced frequency dependencies. One reason for this is that in order to define frequency dependence of the superconducting state, the anomalous Green's function $F$ must be studied, while this work is an introduction into the possibly interesting field of engineering of unconventional superconductivity, and discusses only one example. We believe that if the SAW has certain non-symmetric shape in time, it might give unconventional frequency dependence of the Green's functions $D$ (see Eq. (10)). Time-shaping of SAW is even simpler than in space, because the voltage applied to the interdigital transducer can have various pulse shapes, and the laser pulses can also be modified.

## Acknowledgements

I would like to acknowledge useful discussions with Björn Trauzettel, Tero Heikkilä, Manfred Sigrist, and Andrea Cavalleri. Importantly, I would like to express my deep gratitude to all the people who have been working hard to overcome the health crisis in 2020 and 2021.

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
