# Peer review of "Surface-acoustic-wave-induced unconventional superconducting pairing"

_SciPost Physics, doi:SciPost Phys. Core 5, 005 (2022)_

## Round 1 · Referee Report · Anonymous · 2021-10-10

Strengths
1 - Topic of great interest to the scientific community
2 - High experimental relevance
3 - Good layout and clearly written
Weaknesses
1 - Estimation of relevant experimental parameters would be desirable
Report
The submission studies a topic of high relevance to the physical community. It is written very well and guides the reader nicely through the different sections. As an experimental physicist I cannot judge the validity of the theoretical calculations but from an experimental point of view the submission describes an interesting approach to induce and tune unconventional superconductivity by applying surface acoustic waves. To support the implementation of the proposed technique, it would be very helpful if the most important experimental parameters could be estimated by the author for a suitable candidate material system, for example the SAW power required for the electron-electron interaction to overcome the Coulomb interaction as a prerequisite of inducing unconventional superconductivity.
I am a bit torn if the submission fulfills the high expectations for a SciPost publication, but if the minor adaptations are made, I overall support the publication.
Requested changes
1 - Typo on page 2: "specilfying" instead of "specifying"
2 - Provide estimation of relevant experimental parameters to implement the proposed technique
Author: Viktoriia Kornich on 2021-11-12 [id 1936]
(in reply to Report 1 on 2021-10-10)
(1) Referee 1 wrote: The submission studies a topic of high relevance to the physical community. It is written very well and guides the reader nicely through the different sections.
Response: I would like to thank the referee for high evaluation of the relevance of my work and the clarity of the manuscript.
(2) Referee 1 wrote: As an experimental physicist I cannot judge the validity of the theoretical calculations but from an experimental point of view the submission describes an interesting approach to induce and tune unconventional superconductivity by applying surface acoustic waves.
Response: I am happy that an experimentalist considers my proposal as being interesting, because this somewhat characterizes its adequacy towards real implementation.
(3) Referee 1 wrote: To support the implementation of the proposed technique, it would be very helpful if the most important experimental parameters could be estimated by the author for a suitable candidate material system, for example the SAW power required for the electron-electron interaction to overcome the Coulomb interaction as a prerequisite of inducing unconventional superconductivity.
Response: First of all, I would like to thank the referee for pointing out that this work is not about the actual induction of an unconventional superconducting state, but can serve as a prerequisite of inducing it, because it considers possible induced symmetries of electron pairing. To compare the effects of Coulomb interaction and wave-mediated electron-electron interaction, we need to compare screened Coulomb potential $V_c(k)\propto4\pi e^2/[\epsilon_r(k^2+k_s^2)]$, where $\epsilon_r$ is the static dielectric constant and $k_s$ comes from screening, and the wave-induced electron-electron interaction, $W^K_{\omega_1,\omega_2,{\bf k}_1, {\bf k_2}}$, which is described in Eq. (10) of the manuscript. The form of them is different, taking into account that in general $W^K$ depends on four variables. Therefore direct comparison is not possible in general without significant additional calculations similar to Ref. 36 of the manuscript, where the gap equation was decoupled into different contributions and effective constant for phonon-mediated e-e interaction and Coulomb pseudopotential were derived. Similar derivation was performed e.g. in A. S. Alexandrov, Phys. Rev. B 77, 094502 (2008) for layered structures of cuprates. While I am not sure about the physical effect reported in this reference (I guess, it is still under debate), I think, the algorithm of the derivation of the effective interaction constants and their comparison can be applied to my future work. Moreover, the screening of Coulomb interaction can also be varied in a way to reduce the necessary part of it. To clarify this, the solution of Poisson equation for the electric potential might be needed. Just to demonstrate that it is a question worth of a separate study, see e.g. Phys. Rev. Lett. 127, 187001 (2021), where authors consider only phonon-induced superconductivity and mention Coulomb effects as a part of discussion. I hope to proceed working on this topic and work out this question in detail. If I try to state anything concrete right now, it would be speculation and might confuse future research in this direction taking into account wide variety of materials and gated structures that can be fabricated nowadays. Actually, as was pointed out by the Referee 2, Coulomb interaction is not necessarily absolutely bad for unconventional superconductivity, please, see point 4 of my response to Referee 2.
Changes made: I have added a short description of the above discussion into the manuscript, see Discussion and Conclusions, shown in blue. As this question is directly related to the effect of Coulomb interaction mentioned by Referee 2, please, see also the response to Referee 2, point 4.
(4) Referee 1 wrote: I am a bit torn if the submission fulfills the high expectations for a SciPost publication, but if the minor adaptations are made, I overall support the publication.
Response: I believe this work has necessary importance within the field, because I had three invited talks and two contributed conference talks about it within the last year. I would like to thank the referee for careful reading, an overall positive evaluation, and constructive remarks and questions.
Author: Viktoriia Kornich on 2021-11-12 [id 1937]
(in reply to Report 2 on 2021-10-25)(1) Referee 2 wrote: In this paper, the author discusses how acoustic waves, quite analogous to phonons, can mediate superconductivity. It is pointed out that since frequency and momentum dependences of an acoustic wave can be engineered, this can be used to obtain unconventional superconductivity, such as odd-parity even-frequency superconductivity. The Rayleigh surface acoustic wave, with its spatial odd-parity coupling to electrons, is presented as a prime example.
Response: I would like to point out that this paper is about superconducting electron pairing, i.e. $\psi\psi$ or $\psi^\dagger\psi^\dagger$, and not about actual superconducting state. This point is especially underlined in the end of Introduction. This is important, because induction of a superconducting state is a very complicated problem, which will require to study other electron-electron interactions (e.g., Coulomb), external conditions (e.g., temperature, pressure, gating of the sample), possible structures and/or materials that can host such superconductivity, non-equilibrium processes that will occur as a result of pumping of the system, co-existence of different orders (e.g., spin-triplet and spin-singlet), and probably other questions. All these points are mentioned in the manuscript, and mainly outlined in the Introduction and Discussion and Conclusions. The referee mentions other possible questions, that can arise, further in the report. Thus, the study of the SAW-induced superconducting phase transition will probably take at least several years with several papers dedicated to different aspects and questions. This particular work is the first step, and I hope to proceed working in this field further.
(2) Referee 2 wrote: The author have certainly made a convincing statement for the motivation to study the acoustic wave mediated superconductivity. Also, this paper should also be recommended for clearly presenting how acoustic wave mediated electron-electron interaction is derived.
Response: I would like to thank the referee for the positive evaluation of my manuscript with respect to presentation of motivation and calculations.
( 3) Referee 2 wrote: However, one cannot help but feel that the discussion presented here is not adequate in the sense that the author does not mention whether the odd-parity coupling between the acoustic wave and superconductivity is sufficient for odd-parity superconductivity or other additional conditions is required.
Response: I assume, the referee meant the coupling between the wave and electrons. As I have mentioned in p. 1 of my response, this work does not make statement on whether the superconductivity will be induced. As the referee is asking specifically about odd-parity superconductivity, I would like to mention that there can be other symmetries of pairing present in the system apart from odd-parity ones. E.g. even-parity spin-singlet, as is underlined at the beginning of Discussion and Conclusions of the manuscript. The co-existence of different orders and possible dominance of one order over the other is a question for a separate detailed study investigating superconducting state of matter.
Changes made: I have rewritten the first paragraph of Discussion and Conclusions to make this statement clearer.
(4) Referee 2 wrote: Such problem has been treated extensively in the recent literature (e.g. Phys. Rev. B 90, 184512; Phys. Rev. Lett. 115, 207002; Phys. Rev. B 93, 134512; Phys. Rev. Materials 4, 034202) on superconductivity mediated by odd-parity phonons. This problem is all the more relevant to the present paper as all the listed papers conclude that the additional short-range repulsion is required to stabilize the odd-parity superconductivity, which cannot be stabilized by phonons alone (e.g. Phys. Rev. B 93, 174509);
Response: I would like to thank the referee for pointing out the effect occurring due to electrons' short-range interactions that most likely will play a role, once the condition for the superconducting transition and the co-existence of different orders there are studied. It was my pleasure to study the suggested literature and enrich the discussion of my manuscript with these effects.
Changes made: I have introduced the suggested literature with the explanation of the reported effects of short-range electron-electron interactions on the balance between odd- and even-parity superconducting states in Discussion and Conclusions, shown in blue.
(5) Referee 2 wrote: indeed the recent discussion on the superconductivity in STO through odd-parity phonons alone have all found that this mechanism leads to an even-parity superconductivity (Phys. Rev. B 101, 174501, arXiv:2109.13207, arXiv:2110.03710).
Response: I thank the referee for these examples of theoretical studies of effects in STO due to softening of transversal phonon mode near the critical point. These examples are related to the specific material and many other conditions, therefore I do not think, we can draw a general conclusion based on them. For more general discussion, please, see point 6 of my response.
Changes made: I have cited these papers in Discussion and Conclusions after discussing short-range effects, shown in blue.
(6) Referee 2 wrote: How surface acoustic waves would compare to odd-parity phonons in this respect is a problem of great interest that the author simply failed to acknowledge at all.
Response: I am really grateful to the referee for formulating this question, because I realized that I need to expand and slightly rewrite the related discussion in the paper. My answer is the following. The basis for this manuscript is that superconductivity appears due to symmetry breaking. Conventional superconductivity is a consequence of gauge symmetry breaking. Thus, the expectation is that breaking of other symmetries will most likely allow for other superconducting states, which are regarded as unconventional. The externally applied SAW is a perfect tool for breaking symmetries, because it breaks at least spatial and time translation symmetries of the crystal. In this work, I do not focus of time-related symmetries, but more on spatial symmetry. This breaking of symmetry is not characteristic to thermal phonons, because they are induced by the crystal itself. However, I believe that if the crystal is being deformed or there is a soft mode as discussed in the references suggested by the referee, there are additional symmetries broken, and therefore upon certain conditions (short-range coupling or others), they can lead to unconventional superconductivity (even-, odd- or mixed-parity). My personal opinion is that thermal phonons (even and/or odd) cannot lead to unconventional superconductivity without any symmetry breaking in the system. However, some phonons, e.g. soft phonon mode described in the referee's report, are a consequence of symmetry breaking, and thus are naturally associated with it. This is a different case, and I believe that such phonons in the end can bring unconventional superconducting state (even- and/or odd-parity). Following the referee's question, I underlined the role of the symmetry breaking produced in my setup on symmetries of electron pairing and explained the difference to thermal phonons in the manuscript.
Changes made: I have added the discussion about the SAW-induced structure of $V({\bf k})$ in Section 3.1 after Eq. (8). I have updated Section 3.3. I have also underlined the role of symmetry breaking in the Discussion and Conclusions once more. All these updates are shown in blue.
(7) Referee 2 wrote: Also the band structure of electrons goes unmentioned in this paper. But a surface acoustic wave would couple more strongly to the surface bands. The Rashba effect on these bands have been extensively studied and its effect on the competition between the even-parity and odd-parity superconductivity (e.g. Phys. Rev. Lett. 92, 097001) is another possible complication.
Response: This reference is definitely relevant to the topic, but the question is also related to the actual induction of superconductivity and must be considered in the future studies provided the material or the structure of interest is selected. Right now I think that flat-band systems can be the best platform for my proposal: I have mentioned electron band structure and density of states in them in Discussion and Conclusions. There is also a reference related to the study of surface superconductivity in them. I also would like to note that my proposal is not focused on odd-parity superconductors. It is more about general principle of engineering of unconventional superconductivity, which can be even-, odd- or mixed-parity. The demonstration of the general principle is focused on p-wave pairing, because it is simple.
(8) Referee 2 wrote: In short, while this paper points out an interesting new frontier for study of superconductivity, it is quite worryingly unaware of relevant issues that arose in the directly analogous case. At the minimum, the author needs to discuss how the stability of odd-parity superconductivity mediated by surface acoustic waves compare to that mediated by odd-parity superconductivity.
Response: I have introduced relevant discussions and references in the manuscript and would like to thank the referee for extensive recommendations regarding literature, discussion of various effects, and helpful questions and comments that improved my manuscript.

---

## Round 1 · Referee Report · Anonymous · 2021-10-25

Strengths
1. Strong in originality of the proposal
2. Strong in experimental feasibility of the proposal
Weaknesses
1. Near absence of discussion on the condition for surface acoustic wave to induce odd-parity superconductivity , especially in comparison with the analogous case of the superconductivity mediated by odd-parity phonons
2. Lack of discussion on surface electronic states
Report
In this paper, the author discusses how acoustic waves, quite analogous to phonons, can mediate superconductivity. It is pointed out that since frequency and momentum dependences of an acoustic wave can be engineered, this can be used to obtain unconventional superconductivity, such as odd-parity even-frequency superconductivity. The Rayleigh surface acoustic wave, with its spatial odd-parity coupling to electrons, is presented as a prime example.
The author have certainly made a convincing statement for the motivation to study the acoustic wave mediated superconductivity. Also, this paper should also be recommended for clearly presenting how acoustic wave mediated electron-electron interaction is derived.
However, one cannot help but feel that the discussion presented here is not adequate in the sense that the author does not mention whether the odd-parity coupling between the acoustic wave and superconductivity is sufficient for odd-parity superconductivity or other additional conditions is required. Such problem has been treated extensively in the recent literature (e.g. Phys. Rev. B 90, 184512; Phys. Rev. Lett. 115, 207002; Phys. Rev. B 93, 134512; Phys. Rev. Materials 4, 034202) on superconductivity mediated by odd-parity phonons. This problem is all the more relevant to the present paper as all the listed papers conclude that the additional short-range repulsion is required to stabilize the odd-parity superconductivity, which cannot be stabilized by phonons alone (e.g. Phys. Rev. B 93, 174509); indeed the recent discussion on the superconductivity in STO through odd-parity phonons alone have all found that this mechanism leads to an even-parity superconductivity (Phys. Rev. B 101, 174501, arXiv:2109.13207, arXiv:2110.03710). How surface acoustic waves would compare to odd-parity phonons in this respect is a problem of great interest that the author simply failed to acknowledge at all.
Also the band structure of electrons goes unmentioned in this paper. But a surface acoustic wave would couple more strongly to the surface bands. The Rashba effect on these bands have been extensively studied and its effect on the competition between the even-parity and odd-parity superconductivity (e.g. Phys. Rev. Lett. 92, 097001) is another possible complication.
In short, while this paper points out an interesting new frontier for study of superconductivity, it is quite worryingly unaware of relevant issues that arose in the directly analogous case. At the minimum, the author needs to discuss how the stability of odd-parity superconductivity mediated by surface acoustic waves compare to that mediated by odd-parity superconductivity.

---

## Round 2 · Referee Report · Anonymous (Referee 2) · 2021-12-10

Report

The main merit of this paper lies in stimulating experimental research, as it shows how pairing interaction in the odd-parity pairing channel can be induced by driving surface acoustic wave even when there is no low energy odd-parity phonon present in the material. The author's aim is limited, however, by the fact that it neither present specific material candidates nor seeks to analyze competition between different possible superconducting phases. In that sense, the paper seems more suitable to SciPost Physics Core.

---

## Round 2 · Author Response

I would like to thank the Editor and the referees for providing insightful and helpful comments and questions. I have introduced all requested discussions and replied to the questions (see also my replies to the referees).

Yours sincerely,
Viktoriia Kornich

---

## Editorial Decision

published